# SD-YOLO: An Attention Mechanism Guided YOLO Network for Ship Detection

1st Yunze Zhang
*College of*
*Marine Electrical Engineering*
*Dalian Maritime University*
Dalian, China
zhyz96@163.com

2nd Li-Ying Hao*
*College of*
*Marine Electrical Engineering*
*Dalian Maritime University*
Dalian, China
haoliying_0305@163.com

3rd Yifei Li
*College of*
*Marine Electrical Engineering*
*Dalian Maritime University*
Dalian, China
a19818924273@163.com

*Abstract*—Synthetic Aperture Radar (SAR) is extensively used for vessel detection due to its capability to produce high-resolution images. However, detecting vessels in SAR imagery remains challenging because of the small object sizes and reduced resolution caused by long observation distances, often resulting in high miss-detection rates. To address this issue, this paper introduces a novel detection model—ship detection YOLO (SD-YOLO), which improves small object detection accuracy while maintaining real-time performance. Specifically, we enhance the C3 module of YOLOv5 by incorporating Coordinate Attention (CA) and a bottleneck mechanism, forming the CB-C3 module. Additionally, to increase detection precision and training efficiency, we implement the $\alpha$-IoU loss function, which better constrains detection bounding boxes, enabling the model to locate ships more accurately. We also redesign YOLOv5's neck layer using a Bi-directional Feature Pyramid Network (BiFPN) to optimize multi-scale feature fusion. Experiments on several public SAR datasets demonstrate that SD-YOLO achieves an Average Precision (AP) of 96.1% on the SAR ship detection data-set (SSDD) and 73.2% on the large-scale SAR ship detection data-set (LS-SSDD), representing improvements of 2.7% and 7.9%, respectively. Furthermore, SD-YOLO is more lightweight than other mainstream algorithms, with only 6.79M.

*Index Terms*—ship detection; YOLOv5; coordinate attention; $\alpha$-IoU; BiFPN

## I. INTRODUCTION

Ship detection is crucial for a wide range of applications, including search and rescue operations, maritime traffic control, and automated fisheries management [1]. Although numerous ship detection methods exist, the marine environment presents unique challenges, such as varying weather conditions, ocean waves, and other unpredictable natural factors, which complicate detection efforts. In this context, Synthetic Aperture Radar (SAR) offers a significant advantage due to its ability to capture high-resolution images over large areas, regardless of weather or lighting conditions. SAR is particularly effective in detecting vessels on the ocean surface [2], making accurate and real-time SAR-based ship detection results essential.

Traditional ship detection models rely on manual feature extraction and prior knowledge, making it challenging to effectively capture target characteristics. Consequently, these models often lack robustness and generalization capabilities. In contrast, deep learning has made significant advancements in computer vision, particularly in areas such as semantic segmentation [3], object recognition [4], and object detection [5]. By learning key features from large-scale datasets, deep learning algorithms have demonstrated superior accuracy in object detection and exhibit strong robustness.

Deep learning-based detection methods are typically classified into two categories: two-stage and single-stage algorithms [6]. Two-stage detectors include models such as Region-based Convolutional Neural Networks (R-CNN) [7], Fast R-CNN [8], and Faster R-CNN [9]. R-CNN generates region proposals and normalizes them before performing classification. Fast R-CNN and Faster R-CNN compute features at a single scale, striking a balance between accuracy and processing efficiency. However, these models are generally too slow to meet the real-time processing requirements of embedded systems [10]. To address the demand for faster detection, single-stage models have been developed. These detectors, such as Corner-Net, You Only Look Once (YOLO) [6], and Single Shot MultiBox Detector (SSD) [11], bypass the region proposal stage and directly perform detection on densely sampled positions. YOLO, in particular, utilizes a feed-forward convolutional network for both object localization and classification [12], making it a popular choice for ship detection tasks.

Despite its strengths, YOLO does not effectively account for the spatial relationships between objects in input images [13], which hampers its detection performance, particularly for small vessels. To address this issue, various studies have proposed improvements to the YOLOv5 model in recent years. For instance, Zheng et al. [14] improved the loss function by introducing Distance Intersection over Union (DIoU) and accounting for the Euclidean distance between actual center points, leading to better detection accuracy. Likewise, Sun et al. [15] used a combination of top-down and bottom-

* Corresponding author.

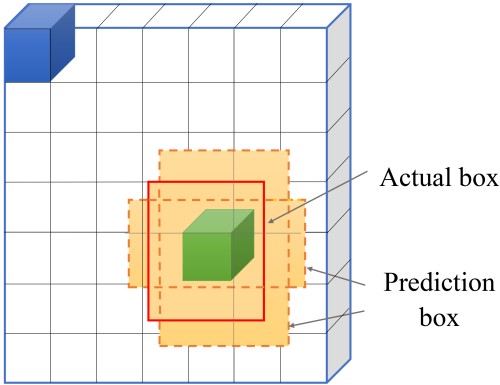

Fig. 1. YOLO Object Detection Process.

### A. YOLO

YOLO is one of the most advanced real-time object detection systems available today and has been widely adopted in ship detection [6]. Unlike the R-CNN series [5], YOLO approaches object detection as a regression problem. The YOLO detection process is depicted in Figure 1. To generate a fixed-size output after applying convolutions to the entire image, the input is resized to $416 \times 416$ pixels. This resized image is then divided into a grid of cells, each tasked with detecting potential bounding boxes. Each cell provides five parameters: $w, h, x, y$ and $confidence$ [6]. Here, $(w, h)$ represent the dimensions of the target's bounding box, while $(x, y)$ denote its coordinates. The $confidence$ score reflects both the likelihood of an object being present within the cell and the accuracy of the predicted bounding box. In cases where multiple bounding boxes detect the same object, YOLO uses Non-Maximum Suppression (NMS) to select the most accurate one.

The input layer serves as the entry point for the object detection model, where adaptive image resizing is applied to improve generalization across various datasets. After preprocessing, the images are processed by a convolutional neural network (CNN). The backbone network, which forms the core of the model, is built upon architectures like AlexNet, ResNet, VGGNet, and GoogLeNet [12]. YOLOv5s introduces several architectural enhancements over YOLOv4, including the addition of the Focus module in its backbone, which is not present in YOLOv4. Furthermore, YOLOv5s incorporates two distinct versions of the Cross Stage Partial Network (CSPNet), distinguishing it further from its predecessor. In the neck layer, YOLOv5s utilizes the Path Aggregation Network (PANet) alongside CSPNet to enhance feature fusion [10]. Finally, in the head module, YOLOv5s replaces DIoU-NMS with Weighted NMS, resulting in a slight performance improvement without increasing computational complexity.

### B. Ship detection method based on remote sensing images

The use of remote sensing technology for ship detection has been a critical area of research in both military and civilian applications [22]. However, Synthetic Aperture Radar (SAR)-based ship detection continues to face challenges, including strong target scattering, multi-scale variations, and background interference, which reduce detection accuracy. To address these issues, innovative solutions like NAS-YOLOX have emerged, combining Neural Architecture Search with a multi-scale attention mechanism to enhance feature extraction and fusion, thereby improving detection precision [22]. In addition to SAR-based methods, researchers have explored alternative approaches to enhance object detection performance in remote sensing images (RSI). For instance, CamoNet, an object camouflage network, introduces imperceptible perturbations to deceive CNN-based detectors, improving detection capabilities [23]. Moreover, efforts have been made to integrate low-precision floating-point algorithms with quantized neural

up pathways to achieve more thorough feature fusion. By linearizing the equations and accurately identifying vessel position coordinates, they further increased the model's precision. While the aforementioned algorithms improve feature extraction network architectures, they fail to effectively integrate contextual information in the final model layers, which is crucial for accurate predictions [14], [15].

Moreover, training deep learning models for target detection requires large datasets. The primary datasets used for synthetic aperture radar (SAR) ship detection are the SAR Ship Detection Dataset (SSDD) [16] and the Large-Scale SAR Ship Detection Dataset (LS-SSDD) [17]. Key differences exist between SAR and natural images: SAR images are single-channel, sparsely distributed, and contain relatively few ships. As a result, when generic object detection models are applied directly to SAR ship detection tasks, their performance tends to degrade [18]. This underscores the urgent need for a highly accurate and robust SAR-specific ship detection model.

In this study, considering the unique characteristics of SAR images, we leverage the speed of the YOLOv5 algorithm to improve ship detection accuracy. The key contributions of this work are outlined as follows:

1) Inspired by the coordinate attention mechanism, we developed a novel feature enhancement module called CB-C3. Unlike previous approaches [19], CB-C3 emphasizes foreground information in input images, significantly improving the model's detection performance, particularly for small vessels.
2) The neck layer integrates a Bidirectional Feature Pyramid Network (BiFPN), which is more efficient than YOLOv5 [12] at extracting and fusing multi-scale feature information, leading to improved accuracy in ship detection.
3) The loss function is enhanced with $\alpha$-IoU, which extends the traditional IoU loss into a power IoU loss by adjusting the $\alpha$ parameter. Unlike previous methods [20], [21], $\alpha$-IoU allows the detector to achieve multi-level bounding box regression when training on different SAR ship datasets.

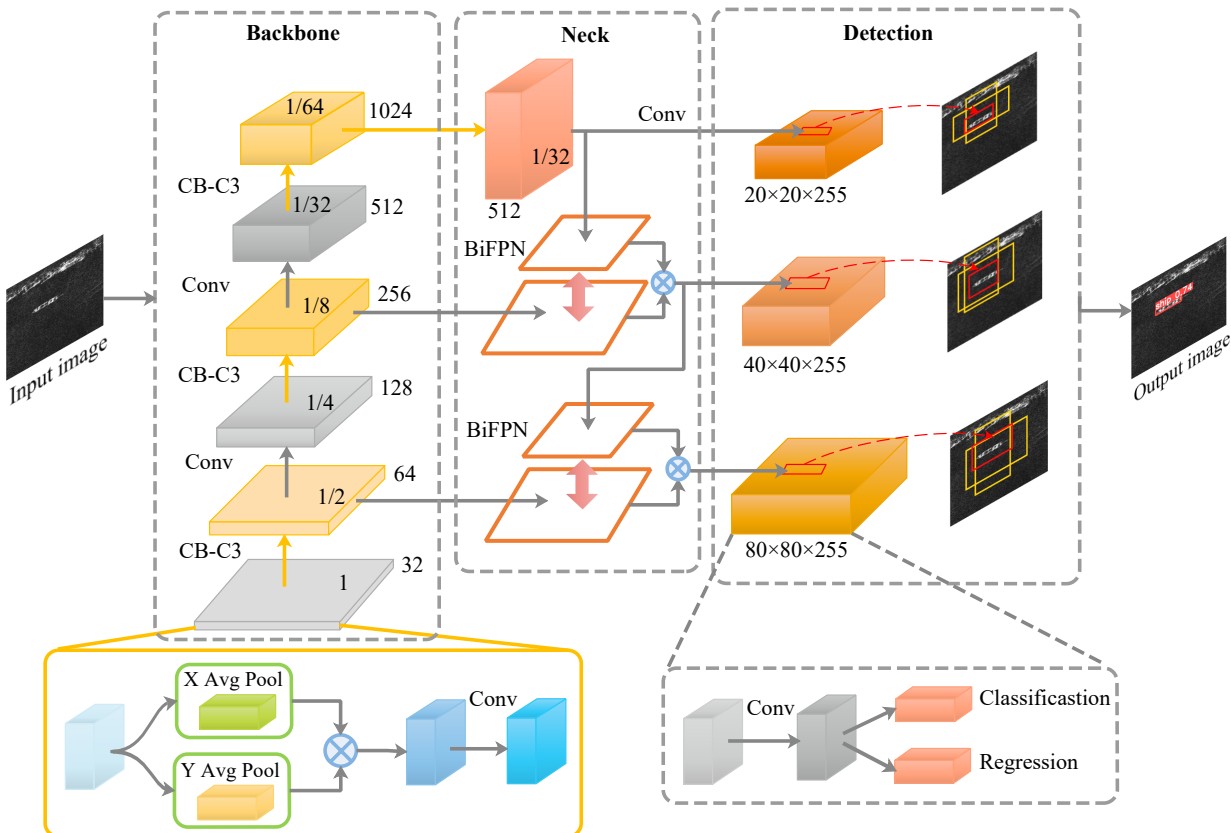

Fig. 2. Overall Architecture of the Proposed Model.

networks, enabling more efficient processing of deep learning models for ship detection, such as in satellite image-based semantic segmentation tasks [24]. Further research has focused on developing custom convolutional neural networks, specifically training SAR ship detectors from scratch to meet unique requirements and enhance detection accuracy [25]. To address class imbalance in SAR ship detection datasets, interpretable evidence learning has been proposed as a solution for learning from biased samples [26]. High-resolution feature generators have also been developed to improve the detection of small ships in optical remote sensing images, tackling ongoing challenges in this domain [27]. Optimized models, such as YOLO-OSD, leverage hybrid data-driven approaches to enhance ship localization in multi-resolution SAR images [28]. Additionally, unsupervised domain adaptation techniques based on cross-domain feature interaction and balanced data contribution have been introduced to improve detection performance across diverse datasets [29]. Collectively, these advancements underscore a range of methods leveraging remote sensing technology to enhance ship detection accuracy.

## III. METHODOLOGY

### A. Method Overview

The SD-YOLO framework, illustrated in Figure 2, incorporates CSPDarknet53 as its backbone, similar to YOLOv5, to extract multi-scale features from SAR images. To effectively capture contextual information, the framework introduces a novel CB-C3 module during the feature extraction stage. For multi-scale feature fusion, the high-performance BiFPN is employed in the neck layer. Detailed descriptions of these components are provided in Sections B and C. Given the importance of bounding box regression in determining the model's detection accuracy, we introduce the $\alpha$-IoU, a power IoU loss family, during the prediction and classification stages. The $\alpha$-IoU generalizes the loss function through a unified power parameter. A comprehensive explanation of this loss function is provided in Section D.

### B. CB-C3 Module

Attention mechanisms are commonly used in object detection tasks to help models focus on key areas, particularly the foreground of the image. Since the introduction of attention techniques like squeeze-and-excitation networks [30], efficient channel attention [31], and the convolutional block attention module [32], the detection performance of YOLO models has seen significant improvement. However, the increased computational overhead associated with these mechanisms has limited their use, particularly in networks with lower computational capacity. To address this issue, we propose the Coordinate Attention (CA) mechanism, designed to enhance the network's accuracy without adding substantial computational burden.

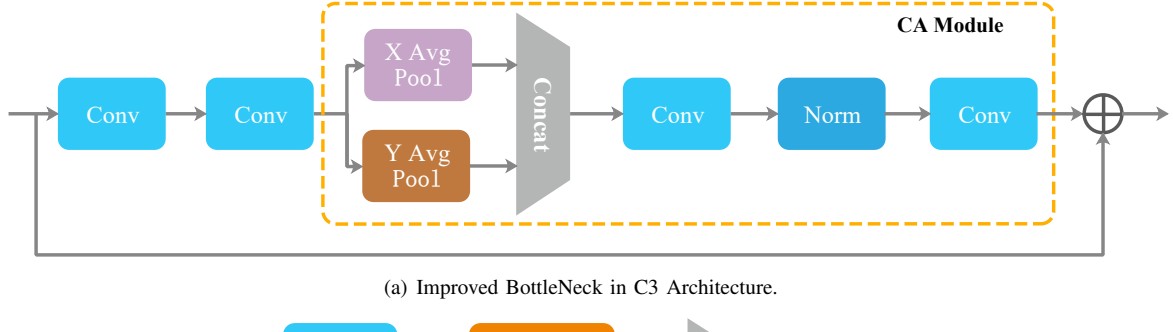

(a) Improved BottleNeck in C3 Architecture.

(b) C3 Module Architecture.

Fig. 3. Framework of the Improved CB-C3.

The Coordinate Attention (CA) mechanism is designed for simplicity and adaptability, making it suitable for use across various neural network architectures. A comprehensive discussion of CA can be found in [33], which highlights its superior performance compared to traditional attention mechanisms. CA improves the efficiency of information transmission by guiding convolutional neural networks to concentrate on important coordinates while filtering out irrelevant information.

The C3 module in YOLOv5 offers fewer parameters and faster inference speed; however, its simple convolutional layer structure leads to poor performance in detecting small objects. To overcome this limitation, we propose the CB-C3 module, with its detailed structure shown in Figure 3. In the original C3 module, the bottleneck structure primarily relies on $3 \times 3$ convolutions for feature extraction, which limits the receptive field. To address this, we integrated average pooling within the CA mechanism to enhance YOLOv5's ability to capture global information more effectively.

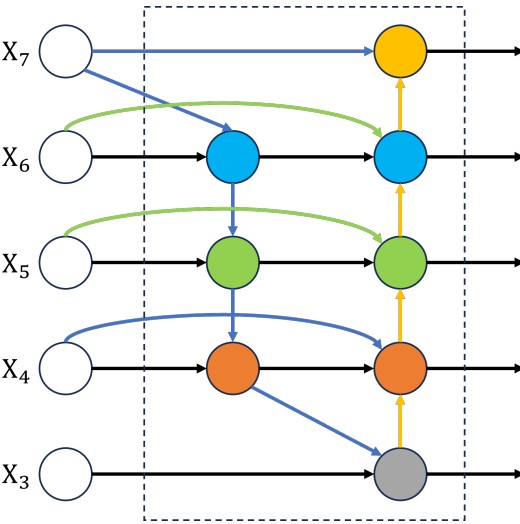

Fig. 4. Bidirectional Feature Pyramid Network Architecture.

### C. Feature Fusion Network Optimization

In the YOLOv5 architecture, the Concat operation serves as the feature fusion module in the neck layer, concatenating input tensors along a specified dimension to enable higher-dimensional data processing. While this method effectively merges multiple tensors, it has drawbacks, such as increased computational load and the risk of dimensionality explosion. To improve the multi-scale fusion of ship features extracted from the backbone, we redesigned YOLOv5's neck layer by incorporating the BiFPN.

Feature fusion networks are crucial for capturing the contextual significance of input images. The BiFPN, introduced by Mingxing Tan in July 2020, offers two key advantages [18]. First, it enhances the efficiency of feature fusion compared to the original FPN by integrating a downsampling path that transitions from high to low resolutions. Unlike PANet, BiFPN is both more efficient and lightweight, as it removes nodes that rely solely on single inputs [34]. Second, BiFPN introduces skip connections between inputs and outputs, enabling the network to more effectively combine low-level and high-level features. Stacking multiple BiFPN modules allows the model to progressively refine feature fusion as its depth increases. The structure of the BiFPN module is illustrated in Figure 4, where $X_i$ represents the feature from the $i$-th input.

### D. Loss Function Optimization

Intersection over Union (IoU) is a critical metric for evaluating object detection algorithms and calculating Average Precision (AP). IoU measures the ratio of the overlapping area between the predicted bounding box and the ground

truth box to their union. However, when there is no overlap between the predicted and actual boxes, the gradient becomes zero, rendering the traditional IoU calculation ineffective. To address this issue, improved IoU methods, such as Generalized IoU (GIoU) [5], have been developed. GIoU adds a penalty term to the loss function, allowing the model to continue learning even in cases where the predicted and ground truth boxes do not overlap. Further advancements, including DIoU [35] and Complete IoU (CIoU) [21], incorporate additional factors such as aspect ratio. These methods are designed to handle situations where one bounding box fully overlaps another, further reducing the impact of the gradient vanishing problem.

However, there are large differences between SAR images and optical images, for example, the ship targets in SAR images are smaller and more widely distributed, and the above algorithms have certain limitations when detecting ships in SAR images. In order to improve the prosperity of the algorithm, this paper employs a new loss function called $\alpha$-IoU [36]. The general IoU loss is defined as $L_{IoU} = 1 - IoU$. The $\alpha$-IoU first use the Box-Cox transformation and generalize the IoU loss to an $\alpha$-IoU loss:

$$L_{\alpha-IoU} = 1 - IoU^{\alpha}, \alpha > 0, \tag{1}$$

$$L_{\alpha-DIoU} = 1 - IoU^{\alpha} + \frac{\rho^{2\alpha(b, b^{gt})}}{c^{2\alpha}}, \alpha > 0, \tag{2}$$

furthermore, the YOLOv5 IoU loss function can be generalized as:

$$L_{\alpha-GIoU} = 1 - IoU^{\alpha} + (\frac{|C\backslash(b \cap b^{gt})|}{|C|})^{\alpha}, \alpha > 0, \tag{3}$$

where b, $b^{gt}$ denote the central of anchor box and target box respectively. $\rho$, defined as $\|b - b^{gt}\|_2$, quantifies the Euclidean distance. C denotes the smallest bounding region that includes both the ground truth box and the predicted box. By adjusting the $\alpha$ parameter, the current IoU-based loss can be extended into a new family of IoU losses. This new family comprises the IoU term and an additional regularization term governed by $\alpha$.

## IV. EXPERIMENTAL AND DISCUSSION

### A. Datasets

To assess the performance of our proposed method, we employed the SSDD [16] and LS-SSDD [17] datasets. SSDD [16] is an essential publicly available dataset for SAR ship detection, containing 1,160 SAR images and 2,456 ship instances. LS-SSDD [17], introduced by Li et al. in 2020, addresses the issue of limited data volume in SSDD [16] Table I. And we converted both datasets into YOLO and COCO formats.

The network is trained on a Linux system equipped with an NVIDIA GTX3090 GPU with 24GB of memory and an Intel(R) Core(TM) i9-10940X CPU. Various maritime target detection algorithms were compared and evaluated based on

TABLE I
SSDD AND LS-SSDD DISTRIBUTION

| Dataset | - | Number of Samples |
|---|---|---|
| SSDD | Training Set | 812 |
| | Validation Set | 116 |
| | Testing Set | 232 |
| | Total | 1160 |
| LS-SSDD | Training Set | 6300 |
| | Validation Set | 900 |
| | Testing Set | 1800 |
| | Total | 9000 |

speed, accuracy, and hardware requirements. Multiple experimental results validated the effectiveness of the proposed model.

### B. Evaluation Metrics

The formulas for calculating Precision, Recall, and IoU are as follows with True Positive (TP), False Positive (FP) and False Negative (FN):

$$Precision = \frac{TP}{TP + FP}, \tag{4}$$

$$Recall = \frac{TP}{TP + FN}, \tag{5}$$

$$IoU = \frac{S_{over}}{S_{union}}, \tag{6}$$

where $S_{over}$ refers to the overlapping area between the ship target bounding box and the ground truth, while $S_{union}$ denotes the area of their union. An IoU value exceeding 0.5 means that the predicted bounding box overlaps with at least half of the ground truth, which qualifies as a correct object detection and corresponds to $AP_{50}$. The definitions for $AP_{75}$ and $AP_{50:95}$ are based on a similar concept.

### C. Ablation Study and Analysis

To demonstrate the effect of the newly proposed loss function and innovative network design, ablation studies were performed on both the SSDD [16] and LS-SSDD [17] datasets. For consistency and fairness, the original YOLOv5 method served as the baseline for comparison. Initially, during the optimization of the $\alpha$-loss function, experiments were performed to determine the optimal $\alpha$ values for different datasets, with the goal of maximizing the algorithm's accuracy through gradient descent. Table II presents the experimental results for various $\alpha$ values.

We observed that for LS-SSDD [16], the highest accuracy is achieved when $\alpha$=3. In contrast, for SSDD [17], the optimization is more effective when $\alpha$=5. Consequently, we select the most suitable $\alpha$ value for each detection scenario. To verify the effectiveness of optimizing YOLOv5, we combined three different optimization methods and conducted ablation experiments on the SAR datasets. Table III presents the experimental results across multiple image scenarios.

TABLE II
EXPERIMENTAL RESULTS OF DIFFERENT $\alpha$

| Parameter | SSDD | | | LS-SSDD | | |
|---|---|---|---|---|---|---|
| $\alpha$ | $AP_{50}$ | $AP_{75}$ | $AP_{50:95}$ | $AP_{50}$ | $AP_{75}$ | $AP_{50:95}$ |
| 1 | 93.4 | 66.1 | 60.4 | 65.3 | 14.5 | 26.2 |
| 2 | 94 | 65.3 | 61.2 | 65.2 | 14.4 | 26.9 |
| 3 | 94.8 | 66.6 | 61.2 | **65.5** | **14.9** | **27.3** |
| 4 | 94.7 | 65.5 | 61.3 | 65.1 | 14.7 | 26.1 |
| 5 | **95.1** | **67.2** | **62** | 64.5 | 14 | 24.8 |

It is evident that the algorithm's precision improves, with $AP_{50}$ increasing by 1.3% for SSDD [16] and 6% for LS-SSDD [17]. Furthermore, when YOLOv5 is further optimized using BiFPN and the new loss function, the algorithm's performance is enhanced, achieving $AP_{50}$ values of 96.1% and 73.2%, respectively.

### D. Algorithm Performance Comparison

To assess the overall effectiveness of the proposed SD-YOLO, we compared it with several recent advanced methods. These methods include Cascade R-CNN [37], YOLOX [19], Faster R-CNN [9], Tridentnet [38], and YOLOv4 [20]. Ensuring fairness, all algorithms were tested using the same batch size. The results validate the effectiveness and accuracy of the image processing method, network design, and loss function optimization.

Table IV presents the numerical outcomes obtained by various methods on SSDD [16] and LS-SSDD [17] datasets. The experimental findings across multiple scenarios indicate that this method achieves superior accuracy compared to other existing approaches. During SSDD [16] training, SD-YOLO's $AP_{50}$ exceeds Centripetalnet [39] by 5.7%, and in LS-SSDD [17], it outperforms Sparse R-CNN [40] by 19%. Moreover, our method ranks just behind YOLO-X [19] in FPS performance. Despite this, the approach discussed in this paper maintains a strong advantage in real-time execution. The experimental outcomes under typical scene conditions are illustrated in Fig. 5. For smaller targets, YOLOv5 experiences some missed detections, whereas SD-YOLO shows impressive robustness and a high recognition rate.

### E. Discussion

Based on the experimental data in Table II, we observed that as $\alpha$ increases, the model's accuracy improves to a certain extent. However, the $\alpha$ value should not be increased indiscriminately to enhance detection performance. For LS-SSDD [17], when $\alpha$ exceeds 3, the algorithm's accuracy tends to decrease. We believe that indiscriminately increasing the $\alpha$ value may lead to overfitting issues. Therefore, different $\alpha$ values should be selected for various application scenarios. By analyzing the first and second sets of data in Table III, we discovered an interesting phenomenon: when the CA structure is added to C3, the number of parameters in the algorithm does not increase but actually decreases to some extent. We propose two possible reasons for this. First, compared to traditional parameter matrices, CA introduces a parameter-sharing mechanism that reduces the number of parameters at each position by learning attention from coordinate positions. Second, CA incorporates dimensionality reduction operations, which effectively represent coordinate information and reduce the dimensionality of input features, thereby decreasing the number of parameters. In the comparison experiments, although the YOLOv5 algorithm has fewer parameters than YOLO-X, YOLO-X achieves a higher FPS. We attribute this to the structural differences between the algorithms.

## V. CONCLUSION

In this paper, we present a novel ship detection model that improves detection accuracy with minimal increases in algorithmic complexity. Unlike traditional ship detection methods, our approach incorporates an attention mechanism based on the CB-C3 module, which adjusts the weight distribution of features across different scales using 2D global pooling. To further enhance the model's sensitivity to ship targets, we optimize YOLOv5's neck layer by integrating the BiFPN. Additionally, we introduce the $\alpha$ loss function in SD-YOLO to refine the final classification, taking into account the unique characteristics of ships in remote sensing scenarios. This ensures that the classification process is fast, stable, and precise. Experimental results demonstrate that this architecture outperforms benchmark methods in terms of robustness and accuracy on publicly available SSDD and LS-SSDD datasets.

Key challenges in SAR ship detection include dense target distribution, significant dynamic variations, and scene element occlusion. Our future research will focus on developing robust techniques for detecting densely packed targets at multiple scales.

## ACKNOWLEDGMENT

This work is supported by the National Natural Science Foundation of China (Grant Nos. 52171292, 51939001), the Outstanding Young Talent Program of Dalian (Grant No. 2022RJ05).

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

TABLE III
RESULTS OF ABLATION EXPERIMENTS

| Dataset | CB-C3 | BiFPN | $\alpha$-IoU | AP$_{50}$ | AP$_{75}$ | AP$_{50:95}$ | Paramrs(M) |
|---------|-------|-------|--------------|-----------|-----------|--------------|------------|
| SSDD    |       |       |              | 93.4 | 66.1 | 60.4 | 7.02 |
|         | ✓     |       |              | 94.7 | 67.2 | 62.1 | **6.7** |
|         | ✓     | ✓     |              | 95.9 | 68.1 | 63   | 6.79 |
|         |       | ✓     | ✓            | 95.1 | 67.4 | **63.1** | 7.1 |
|         | ✓     | ✓     | ✓            | **96.1** | **69.2** | 62.3 | 6.79 |
| LS-SSDD |       |       |              | 65.3 | 14.5 | 26.2 | 7.02 |
|         | ✓     |       |              | 71.3 | 15.9 | 27.6 | **6.7** |
|         | ✓     | ✓     |              | 72   | 16.9 | 28   | 6.79 |
|         |       | ✓     | ✓            | 68.3 | 15.7 | 27.3 | 7.1 |
|         | ✓     | ✓     | ✓            | **73.2** | **17.3** | **28.4** | 6.79 |

TABLE IV
RESULTS OF ABLATION EXPERIMENTS

| Dataset | Method | AP$_{50}$ | AP$_{75}$ | AP$_{50:95}$ | Paramrs(M) | FPS | GFLOPs |
|---------|--------|-----------|-----------|--------------|------------|-----|--------|
| SSDD    | Cascade R-CNN [37]  | 93.2 | 68.3 | 61.2 | 69.2 | 44 | 234.3 |
|         | YOLOX [19]          | 93.9 | 62.4 | 56.5 | 9.5 | **81** | 33.4 |
|         | Centripetalnet [39] | 90.4 | 64.2 | 59 | 206.4 | 13 | 1916.1 |
|         | Faster R-CNN [9]    | 94.9 | 67.3 | 60.4 | 41.8 | 56 | 207.5 |
|         | Tridentnet [38]     | 92.7 | 63.3 | 33 | 64.1 | 59 | 822.8 |
|         | Cornernet [41]      | 89.7 | 53.4 | 49.5 | 201.2 | 14 | 1765.6 |
|         | YOLOv4 [20]         | 92.4 | 65.2 | 41 | 8.5 | 66 | 20.4 |
|         | Sparse-RCNN [40]    | 93.2 | 66.2 | 61.2 | 105.2 | 41 | 150.2 |
|         | **SD-YOLO(ours)**   | **96.1** | **69.2** | **62.3** | **6.79** | 76 | **15.4** |
| LS-SSDD | Cascade R-CNN [37]  | 64.3 | 15.1 | 26.4 | 69.2 | 41 | 234.3 |
|         | YOLOX [19]          | 65.9 | 14.2 | 25.9 | 9.5 | **80** | 33.4 |
|         | Centripetalnet [39] | 59.2 | 11.2 | 19.4 | 206.4 | 11 | 1916.1 |
|         | Faster R-CNN [9]    | 67.7 | 15.1 | 26.5 | 41.8 | 54 | 207.5 |
|         | Tridentnet [38]     | 66.9 | 15.2 | 26.6 | 64.1 | 53 | 822.8 |
|         | Cornernet [41]      | 54.8 | 12.1 | 19.3 | 201.2 | 9 | 1765.6 |
|         | YOLOv4 [20]         | 65.5 | 14.8 | 26.4 | 8.5 | 65 | 20.4 |
|         | Sparse-RCNN [40]    | 54.2 | 13.2 | 19.4 | 105.2 | 37 | 150.2 |
|         | **SD-YOLO(ours)**   | **73.2** | **17.3** | **28.4** | **6.79** | 74 | **15.4** |

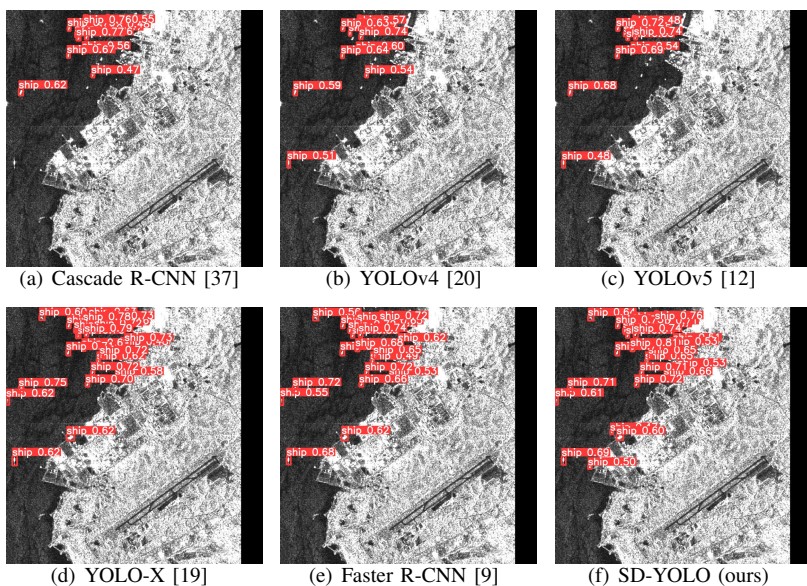

(a) Cascade R-CNN [37]  (b) YOLOv4 [20]  (c) YOLOv5 [12]

(d) YOLO-X [19]  (e) Faster R-CNN [9]  (f) SD-YOLO (ours)

Fig. 5. Comparison Results of the Proposed Method and Other Methods.

[3] S. Zhang and C. Zhang, "Modified u-net for plant diseased leaf image segmentation," *Computers and Electronics in Agriculture*, vol. 204, p. 107511, 2023.

[4] R. P. Babadian, K. Faez, M. Amiri, and E. Falotico, "Fusion of tactile and visual information in deep learning models for object recognition," *Information Fusion*, vol. 92, pp. 313–325, 2023.

[5] X. Zhao, J. Xiao, B. Zhang, Q. Zhang, and A.-N. Waleed, "Weight-guided loss for long-tailed object detection and instance segmentation," *Signal Processing: Image Communication*, vol. 110, p. 116874, 2023.

[6] J. Redmon, S. Divvala, R. Girshick, and A. Farhadi, "You only look once: Unified, real-time object detection," in *Proceedings of the IEEE conference on computer vision and pattern recognition*, 2016, pp. 779–788.

[7] Z. Zou, K. Chen, Z. Shi, Y. Guo, and J. Ye, "Object detection in 20 years: A survey," *Proceedings of the IEEE*, vol. 111, no. 3, pp. 257–276, 2023.

[8] R. Girshick, "Fast r-cnn," in *Proceedings of the IEEE international conference on computer vision*, 2015, pp. 1440–1448.

[9] Y. Li, S. Zhang, and W.-Q. Wang, "A lightweight faster r-cnn for ship detection in sar images," *IEEE Geoscience and Remote Sensing Letters*, vol. 19, pp. 1–5, 2020.

[10] X. Xu, M. Zhao, P. Shi, R. Ren, X. He, X. Wei, and H. Yang, "Crack detection and comparison study based on faster r-cnn and mask r-cnn," *Sensors*, vol. 22, no. 3, p. 1215, 2022.

[11] W. Liu, D. Anguelov, D. Erhan, C. Szegedy, S. Reed, C.-Y. Fu, and A. C. Berg, "Ssd: Single shot multibox detector," in *Computer Vision–ECCV 2016: 14th European Conference, Amsterdam, The Netherlands, October 11–14, 2016, Proceedings, Part I 14*. Springer, 2016, pp. 21–37.

[12] X. Zhu, S. Lyu, X. Wang, and Q. Zhao, "Tph-yolov5: Improved yolov5 based on transformer prediction head for object detection on drone-captured scenarios," in *Proceedings of the IEEE/CVF international conference on computer vision*, 2021, pp. 2778–2788.

[13] J. Wang, Y. Chen, Z. Dong, and M. Gao, "Improved yolov5 network for real-time multi-scale traffic sign detection," *Neural Computing and Applications*, vol. 35, no. 10, pp. 7853–7865, 2023.

[14] Z. Zheng, P. Wang, W. Liu, J. Li, R. Ye, and D. Ren, "Distance-iou loss: Faster and better learning for bounding box regression," in *Proceedings of the AAAI conference on artificial intelligence*, vol. 34, no. 07, 2020, pp. 12 993–13 000.

[15] Z. Sun, X. Leng, Y. Lei, B. Xiong, K. Ji, and G. Kuang, "Bifa-yolo: A novel yolo-based method for arbitrary-oriented ship detection in high-resolution sar images," *Remote Sensing*, vol. 13, no. 21, p. 4209, 2021.

[16] T. Zhang, X. Zhang, J. Li, X. Xu, B. Wang, X. Zhan, Y. Xu, X. Ke, T. Zeng, H. Su *et al.*, "Sar ship detection dataset (ssdd): Official release and comprehensive data analysis," *Remote Sensing*, vol. 13, no. 18, p. 3690, 2021.

[17] F. Xu, J. Liu, C. Dong, and X. Wang, "Ship detection in optical remote sensing images based on wavelet transform and multi-level false alarm identification," *Remote Sensing*, vol. 9, no. 10, p. 985, 2017.

[18] Z. Chen, D. Chen, Y. Zhang, X. Cheng, M. Zhang, and C. Wu, "Deep learning for autonomous ship-oriented small ship detection," *Safety Science*, vol. 130, p. 104812, 2020.

[19] Z. Ge, S. Liu, F. Wang, Z. Li, and J. Sun, "Yolox: Exceeding yolo series in 2021," *arXiv preprint arXiv:2107.08430*, 2021.

[20] A. Bochkovskiy, C.-Y. Wang, and H.-Y. M. Liao, "Yolov4: Optimal speed and accuracy of object detection," *arXiv preprint arXiv:2004.10934*, 2020.

[21] Z. Zheng, P. Wang, D. Ren, W. Liu, R. Ye, Q. Hu, and W. Zuo, "Enhancing geometric factors in model learning and inference for object detection and instance segmentation," *IEEE transactions on cybernetics*, vol. 52, no. 8, pp. 8574–8586, 2021.

[22] H. Wang, D. Han, M. Cui, and C. Chen, "Nas-yolox: a sar ship detection using neural architecture search and multi-scale attention," *Connection Science*, vol. 35, no. 1, pp. 1–32, 2023.

[23] Y. Zhou, W. Jiang, X. Jiang, L. Chen, and X. Liu, "Camonet: A target camouflage network for remote sensing images based on adversarial attack," *Remote Sensing*, vol. 15, no. 21, p. 5131, 2023.

[24] C. Gernigon, S.-I. Filip, O. Sentieys, C. Coggiola, and M. Bruno, "Low-precision floating-point for efficient on-board deep neural network processing," in *2023 European Data Handling & Data Processing Conference (EDHPC)*. IEEE, 2023, pp. 1–8.

[25] J. Li, C. Xu, P. Cheng, C. Chi, L. Yu, Z. Yu, and S. Xu, "Training sar ship detectors from scratch with customized convolutional neural network," *IEEE Transactions on Aerospace and Electronic Systems*, 2023.

[26] Y. Liu, G. Yan, F. Ma, Y. Zhou, and F. Zhang, "Sar ship detection based on explainable evidence learning under intra-class imbalance," *IEEE Transactions on Geoscience and Remote Sensing*, 2024.

[27] H. Zhang, S. Wen, Z. Wei, and Z. Chen, "High-resolution feature generator for small ship detection in optical remote sensing images," *IEEE Transactions on Geoscience and Remote Sensing*, 2024.

[28] M. F. Humayun, F. A. Nasir, F. A. Bhatti, M. Tahir, and K. Khurshid, "Yolo-osd: Optimized ship detection and localization in multiresolution sar satellite images using a hybrid data-model centric approach," *IEEE Journal of Selected Topics in Applied Earth Observations and Remote Sensing*, vol. 17, pp. 5345–5363, 2024.

[29] Y. Yang, J. Chen, L. Sun, Z. Zhou, Z. Huang, and B. Wu, "Unsupervised domain-adaptive sar ship detection based on cross-domain feature interaction and data contribution balance," *Remote Sensing*, vol. 16, no. 2, p. 420, 2024.

[30] J. Hu, L. Shen, and G. Sun, "Squeeze-and-excitation networks," in *Proceedings of the IEEE conference on computer vision and pattern recognition*, 2018, pp. 7132–7141.

[31] Q. Wang, B. Wu, P. Zhu, P. Li, W. Zuo, and Q. Hu, "Eca-net: Efficient channel attention for deep convolutional neural networks," in *Proceedings of the IEEE/CVF conference on computer vision and pattern recognition*, 2020, pp. 11 534–11 542.

[32] S. Woo, J. Park, J.-Y. Lee, and I. S. Kweon, "Cbam: Convolutional block attention module," in *Proceedings of the European conference on computer vision (ECCV)*, 2018, pp. 3–19.

[33] Q. Hou, D. Zhou, and J. Feng, "Coordinate attention for efficient mobile network design," in *Proceedings of the IEEE/CVF conference on computer vision and pattern recognition*, 2021, pp. 13 713–13 722.

[34] M. Tan, R. Pang, and Q. V. Le, "Efficientdet: Scalable and efficient object detection," in *Proceedings of the IEEE/CVF conference on computer vision and pattern recognition*, 2020, pp. 10 781–10 790.

[35] D. Yuan, X. Shu, N. Fan, X. Chang, Q. Liu, and Z. He, "Accurate bounding-box regression with distance-iou loss for visual tracking," *Journal of Visual Communication and Image Representation*, vol. 83, p. 103428, 2022.

[36] H. Li, Q. Zhou, Y. Mao, B. Zhang, and C. Liu, "Alpha-sganet: A multi-attention-scale feature pyramid network combined with lightweight network based on alpha-iou loss," *Plos one*, vol. 17, no. 10, p. e0276581, 2022.

[37] Z. Cai and N. Vasconcelos, "Cascade r-cnn: Delving into high quality object detection," in *Proceedings of the IEEE conference on computer vision and pattern recognition*, 2018, pp. 6154–6162.

[38] Y. Li, Y. Chen, N. Wang, and Z. Zhang, "Scale-aware trident networks for object detection," in *Proceedings of the IEEE/CVF international conference on computer vision*, 2019, pp. 6054–6063.

[39] Z. Dong, G. Li, Y. Liao, F. Wang, P. Ren, and C. Qian, "Centripetalnet: Pursuing high-quality keypoint pairs for object detection," in *Proceedings of the IEEE/CVF conference on computer vision and pattern recognition*, 2020, pp. 10 519–10 528.

[40] P. Sun, R. Zhang, Y. Jiang, T. Kong, C. Xu, W. Zhan, M. Tomizuka, L. Li, Z. Yuan, C. Wang *et al.*, "Sparse r-cnn: End-to-end object detection with learnable proposals," in *Proceedings of the IEEE/CVF conference on computer vision and pattern recognition*, 2021, pp. 14 454–14 463.

[41] H. Law and J. Deng, "Cornernet: Detecting objects as paired keypoints," in *Proceedings of the European conference on computer vision (ECCV)*, 2018, pp. 734–750.
