# OpenReview forum: "SD-YOLO: An Attention Mechanism Guided YOLO Network for Ship Detection"
_IEEE.org/ICIST/2024/Conference — IEEE ICIST 2024 Conference Submission_

### Official Review · Reviewer_QjmJ · 2024-08-21
**A new ship detection model is proposed in this paper. This paper is interesting. However, the following comments should be considered in the revision.**

**Rating:** 7
**Confidence:** 3

**Review:**

Question 1:
Please provide further details on how the CB-C3 module specifically enhances the detection of small ships compared to previous methods. How does the increased focus on foreground information translate into improved performance metrics?
Question 2:
Please elaborate on how the BiFPN neck layer improves the extraction and fusion of multi-scale features? What specific advantages does it offer over YOLOv5 in terms of ship object detection accuracy?
Question 3:
How does the α-IoU loss function's adjustment of the parameter α lead to more effective bounding box regression across multiple SAR ship datasets? Can the authors discuss the benefits of this approach compared to existing IoU loss functions and provide examples of its impact on detection performance?

---

### Official Review · Reviewer_XjAW · 2024-08-21
**The paper is logically clear, the simulation results are credible, and it is recommended for publication.**

**Rating:** 8
**Confidence:** 3

**Review:**

This paper proposes SD-YOLO, an enhanced detection model for ship detection in SAR images, which improves detection accuracy for small targets through modified C3 module, α-IoU loss, and BiFPN neck layer, achieving state-of-the-art performance with reduced parameters. The paper is logically clear, the simulation results are credible, and it is recommended for publication. This paper is well organized and contains meaningful results. However, the following comments should be considered in the revision.
1.The SD-YOLO model has shown promising results in ship detection from SAR images. To further demonstrate its versatility, future research could explore extending the model to detect other maritime objects such as buoys, oil spills, or even marine life. This would not only validate the generalizability of the model but also provide valuable tools for a wider range of maritime surveillance and monitoring applications.
2.Although the SD-YOLO model achieves high accuracy on benchmark datasets, deploying it in real-world scenarios where large amounts of labeled data may not be readily available can be challenging. Incorporating transfer learning techniques by pre-training the model on a larger, more diverse dataset and then fine-tuning it on the target dataset with limited annotations could help overcome this issue. This approach would make the model more accessible and applicable in practical applications.
3.The current SD-YOLO model focuses on detecting ships in individual SAR images. However, with the availability of video SAR data, incorporating spatio-temporal information across frames could potentially further improve detection accuracy. For instance, utilizing optical flow or tracking algorithms to model the motion of ships between frames could help reduce false positives and increase robustness to occlusion or noise in individual frames. Additionally, this could enable the model to predict future ship positions, providing valuable insights for maritime traffic management and collision avoidance systems.

---

### Official Review · Reviewer_LYgs · 2024-08-22
**Manuscript Accept**

**Rating:** 7
**Confidence:** 4

**Review:**

This work is novel. Some issues should be considered in the revision.
Can CB-C3 improve the detection performance of the other models except the small ships?
How does the α-IoU loss function constraint detection bounding box improve the detector performance?
What is the main challenge of having a detection in a large-scale SAR ship detection dataset?

---

### Decision · Program_Chairs · 2024-09-08

Accept (Oral)